# Comparison of generic drug prices in Korea and eight high-income countries across four therapeutic classes

Chanmi Park[1], Dong-Sook Kim[2], Inmyung Song[2*]

1 Biomedical Research Center, Korea University Guro Hospital, Seoul, Republic of Korea, 2 College of Nursing and Health, Kongju National University, Gongju, Republic of Korea

⊙ These authors contributed equally to this work.
* inmyungs@gmail.com

## Abstract

### Background

The prices of generic drugs in general are known to be higher in Korea than in other countries. However, it remains unknown whether the price levels of generic drugs in Korea relative to other countries can differ by therapeutic class. Therefore, this study compared the prices of generic drugs in four commonly used drug classes in Korea with those in other high-income countries.

### Methods

Using IQVIA's Pricing Insight data from 2018 to 2022, we calculated the Laspeyres price index for generic drugs in four therapeutic classes (antidiabetic drugs, lipid-modifying agents, antihypertensive drugs, and antibiotics). We selected eight high-income countries, such as Canada, France, Germany, Italy, Japan, Switzerland, the United Kingdom, and the United States, for comparison and Korea as the base country. Price to chemist was used and the currency conversion was based on the exchange rate and the purchasing power parity.

### Results

Prices of generic drugs are lower in all of comparison countries combined than in Korea for lipid-modifying drugs and antihypertensive drugs. For these two drug classes, all countries but the U.S. have the index lower than one. The index for antidiabetic drugs was less than one in all countries except for Canada and the U.S. For antibiotics, all countries but France, Italy, and Japan have the index that is greater than one. Furthermore, the price index for generic antibiotics increased from 2018 to 2022 in all countries but Canada and Japan.

### Conclusion

The prices of generic drugs are higher in Korea than in other high-income countries for lipid-modifying agents and antihypertensive drugs. The prices of generic antibiotics are higher in many comparison countries and have further increased from 2018 to 2021.

**Data availability statement:** This study is based on third-party proprietary data. The data underlying the results presented in this study are available for purchase from IQVIA (https://www.iqvia.com/). The authors did not have any special access privileges that others would not have.

**Funding:** This study was supported by the Ministry of Health and Welfare of Korea (Ref. No. 2023080955F – 00). The funders had no role in study design, data collection and analysis, decision to publish, or preparation of the manuscript.

**Competing interests:** The authors have declared that no competing interests exist.

## Introduction

Containing the same active ingredient, generic drugs are considered the bioequivalent and lower-priced alternatives to the originator brand-name drug [1]. Many countries that are burdened with skyrocketing drug expenditures have implemented measures to promote the use of generic drugs in many countries [2]. Pharmaceutical expenditure in Korea has nearly doubled from 2007 to 2017 [3]. To control pharmaceutical spending, the Korean government has introduced a range of policies to control drug prices [4]. Among them is the 2012 policy that requires the prices of generic drugs to be set at 59.6% of the price of the originator drug and the price of the originator to be at the 70% of its initial price in the first year of patent expiration [5]. The policy further requires the prices of both the originator and generics alike to be set at 53.5% of the initial price of the originator in the second year of patent expiration, eliminating all differences in prices between the originator and generics [5]. However, this across-the-board price cuts of off-patent drugs have been estimated as having only a transient impact on controlling total pharmaceutical spending [6]. In response, the government introduced another measure in July of 2020, which differentiated the prices of generic drugs depending on the quality of the pharmaceutical product and the number of generic drugs available per molecule. As a result of the measure, the price of a generic is determined at the level that ranges from 38.7% to 53.6% of the maximum price of the originator [7]. Whether the price control policy has any influence on the overall price levels of generics remains to be seen.

Efforts have often been made to understand pharmaceutical price levels in a country relative to others [8,9]. According to a study based on the prices of 80 commonly prescribed generic drugs in Korea in 2008, purchasing power parity (PPP)-adjusted drug prices in Korea are higher than those in 15 other countries [10]. Similarly, a 2021 study based on 23 active ingredients in solid form also showed that generic drug prices were higher in Korea than in 15 comparison countries [11]. The study attributed its findings to small differences in price between the originator and generic drugs in Korea [11]. While these previous studies estimated the generally higher prices of generic drugs in Korea by applying commonly used price indices, it remains unknown whether the price levels of generic drugs in Korea relative to other countries can differ by therapeutic class. It is likely that some generic drugs can be more expensive but that others may be cheaper in Korea than in other countries, depending on the number of manufacturers and generics in a drug class. Therefore, the purpose of this study is to bilaterally compare the prices of generic drugs in four therapeutic classes that are commonly used between eight high-income countries and Korea. In addition, this study examines the temporal trend in the price indices for the selected drugs from 2018 to 2022.

## Methods

### Data

This study used IQVIA's Pricing Insight data from 2018 to 2022. IQVIA is a company that specializes in tracking, collecting, and selling data on pharmaceutical prices and sales in major pharmaceutical markets. The IQVIA dataset contains information on prices, formulations, and dosage strengths of prescription drugs by country. The dataset provides several price points, such as ex-manufacturer price excluding sales tax, price to chemist (PTC), and retail price including sales tax. We used PTC as it approximates the sum of the drug price to the payer and patient's copayment for the drug, representing costs to the society. Retail prices are not chosen because they include sales tax. It has been recommended against using retail prices in international drug price comparison because some countries impose sales taxes on pharmaceuticals and the rates vary [9]. Another problem with using retail prices is that they could include profit margins, which are not allowed in Korea. Instead, pharmacists in Korea are paid a dispensing fee for each prescription filled [10]. We extracted prices in all

countries in their national currencies and converted the prices to US dollars (USD) by using the exchange rate and PPP. Consumption data for generic drugs were extracted from data published by the Health Insurance Review and Assessment Service of Korea.

### Study drugs

There are variations in the definition of the generic drug across countries [12,13]. For the purpose of this study, we defined the generic drug as the drug that has the same active ingredient as the originator drug in the same formulation and strength. We selected generic drugs in three therapeutic classes that have been commonly prescribed for the management of chronic diseases: antidiabetic drugs (anatomical therapeutic chemical [ATC] classification: A10), lipid-modifying agents (C10), and antihypertensive drugs (C02–C09). In addition, we selected antibiotics for systemic use (J01). These drug classes were selected because they are considered to have high consumption levels, based on data from an international repository [14].

### Price index

We calculated the Laspeyres price index, a commonly used price index, to compare prices using Korea as the base country [1]. Comparison countries were eight high-income countries, such as Canada, France, Germany, Italy, Japan, Switzerland, the United Kingdom, and the United States. The rationale for the selection is that these are reference countries that the national authority in Korea considers in determining the reimbursement prices for new pharmaceuticals. Furthermore, the authority announced a plan for a new initiative to reduce generic prices by comparing those in Korea with prices in these eight countries [15]. We first compared prices in Korea with those in all comparison countries combined for each drug class. In addition, we bilaterally compared generic prices in Korea with those in each comparison country.

To compute the index, we identified 291 ingredient-formulation-strength combinations that were listed for reimbursement in Korea. These combinations were equivalent to 143 ATC codes. We further extracted all generic drug products that matched the ATC codes from the IQVIA dataset for all comparison countries. All drugs chosen for comparison were listed in Table 1 Supplementary. The Laspeyres price index for comparison country A ($L_A$) is formulated as follows:

$$L_A = \frac{\sum_i^m p_{Ai} \times q_{Ki}}{\sum_i^m p_{Ki} \times q_{Ki}},$$

where $p_{Ai}$ is the price of the $i$th product (i.e., ingredient-formulation-strength combination) in comparison country A, $q_{Ki}$ is the volume of the $i$th product sold in Korea, and $p_{Ki}$ is the price of the $i$th product in Korea.

We used volume weights for Korea, which represent the share of all generics use that is accounted for by each combination of ingredient, formulation, and strength. By applying volume weights to the prices, the index accounts for differences in the sales volume and mix of generic drugs across countries [9]. The index is presented as the ratio of prices in each comparison country to those in the base country. Therefore, the index higher than one indicates that prices in the comparison country are higher than those in Korea. Analysis was conducted using SAS version 9.4 (Cary, NC, USA).

## Results

### All countries combined

When indices are calculated based on PPP-adjusted prices, the prices of generic drugs are lower in all of comparison countries combined than in Korea for antihypertensive drugs and

lipid-modifyres drugs (Table 1). Moreover, the index has decreased from 2.12 in 2018 to 1.83 in 2022 for antidiabetic drugs and from 0.66 in 2018 to 0.57 in 2022 for lipid-modifying agents. By contrast, the index increased for antihypertensive drugs and antibiotics during the same time period, although it fluctuated from year to year.

### Antidiabetic drugs

The index for antidiabetic drugs was less than one in all countries except for Canada and the U.S. (Table 2). The index decreased considerably from 4.79 in 2018 to 3.40 in 2022 for the U.S. and slightly from 0.85 in 2018 to 0.79 in 2022 for the U.K. The index for Japan was low at 0.57 in 2018 and decreased to an even lower level at 0.49 in 2022. For France, Italy, and Switzerland, the index remained fairly stable during the study period. The index for Canada increased from 1.84 in 2018 to 1.95 in 2020 but came back down to 1.89 in 2022. The index for Germany decreased from 0.67 in 2018 to 0.65 in 2020 but went back up to 0.70 in 2022.

### Lipid-modifying agents

For lipid-modifying agents, all countries except for the U.S. have the index lower than one (Table 3). The U.K. had the lowest index at 0.13 in 2018, which further decreased to 0.10 in 2022. From 2018 to 2022, the index decreased substantially from 0.56 to 0.36 for Japan and from 0.59 to 0.45 for Switzerland. In contrast, the index increased substantially from 0.20 to 0.34 for Germany, from 0.26 to 0.36 for Italy, and from 0.37 to 0.45 for France. The index for Canada stayed stable around 0.26 during the study period, whereas the index for the U.S. fluctuated widely.

### Antihypertensive drugs

For antihypertensive drugs, all countries but the U.S. have the index lower than one (Table 4). Some of these countries saw a further decrease in the index from 2018 to 2022. For example, the index decreased from 0.58 to 0.38 for Japan, from 0.64 to 0.57 for Germany, and from 0.49 to 0.44 for Canada. The index was relatively stable for France, Italy, and Switzerland. On the

Table 1. Laspeyres price indices of generic drugs in four therapeutic classes, 2018–2022.

|  | Antidiabetic drugs | Lipid-modifying drugs | Antihypertensive drugs | Antibiotics |
|---|---|---|---|---|
| Based on USD |  |  |  |  |
| 2018 | 2.68 | 0.83 | 1.01 | 2.14 |
| 2019 | 2.88 | 0.84 | 1.15 | 2.21 |
| 2020 | 2.77 | 0.79 | 1.18 | 2.20 |
| 2021 | 2.49 | 0.71 | 1.08 | 2.31 |
| 2022 | 2.71 | 0.75 | 1.35 | 2.50 |
| Based on PPP USD |  |  |  |  |
| 2018 | 2.12 | 0.66 | 0.82 | 1.74 |
| 2019 | 2.17 | 0.66 | 0.89 | 1.75 |
| 2020 | 2.01 | 0.60 | 0.88 | 1.67 |
| 2021 | 1.89 | 0.57 | 0.84 | 1.80 |
| 2022 | 1.83 | 0.57 | 0.95 | 1.79 |

Base country: Korea; PPP: purchasing power parity.

**Table 2. Laspeyres price indices for antidiabetic drugs, 2018–2022.**

|  | Canada | France | Germany | Italy | Japan | Switzerland | U.K. | U.S. |
|---|---|---|---|---|---|---|---|---|
| Based on USD |  |  |  |  |  |  |  |  |
| 2018 | 2.21 | 0.89 | 0.75 | 0.45 | 0.69 | 1.29 | 1.01 | 6.16 |
| 2019 | 2.42 | 0.89 | 0.75 | 0.45 | 0.71 | 1.24 | 0.97 | 6.41 |
| 2020 | 2.52 | 0.89 | 0.76 | 0.46 | 0.70 | 1.31 | 0.97 | 5.92 |
| 2021 | 2.54 | 0.89 | 0.81 | 0.47 | 0.64 | 1.30 | 0.97 | 5.35 |
| 2022 | 2.76 | 0.89 | 0.83 | 0.49 | 0.56 | 1.41 | 1.03 | 5.27 |
| Based on PPP USD |  |  |  |  |  |  |  |  |
| 2018 | 1.84 | 0.78 | 0.67 | 0.44 | 0.57 | 0.83 | 0.85 | 4.79 |
| 2019 | 1.91 | 0.82 | 0.67 | 0.45 | 0.55 | 0.79 | 0.82 | 4.71 |
| 2020 | 1.95 | 0.78 | 0.65 | 0.44 | 0.52 | 0.77 | 0.79 | 4.20 |
| 2021 | 1.92 | 0.78 | 0.69 | 0.45 | 0.51 | 0.80 | 0.77 | 3.99 |
| 2022 | 1.89 | 0.78 | 0.70 | 0.48 | 0.49 | 0.82 | 0.79 | 3.40 |

Base country: Korea; PPP: purchasing power parity.

**Table 3. Laspeyres price indices for lipid-modifying agents, 2018–2022.**

|  | Canada | France | Germany | Italy | Japan | Switzerland | U.K. | U.S. |
|---|---|---|---|---|---|---|---|---|
| Based on USD |  |  |  |  |  |  |  |  |
| 2018 | 0.33 | 0.42 | 0.23 | 0.27 | 0.68 | 0.92 | 0.16 | 2.87 |
| 2019 | 0.34 | 0.42 | 0.25 | 0.28 | 0.69 | 0.94 | 0.14 | 3.21 |
| 2020 | 0.33 | 0.47 | 0.27 | 0.31 | 0.62 | 1.00 | 0.15 | 2.95 |
| 2021 | 0.34 | 0.51 | 0.29 | 0.31 | 0.51 | 0.81 | 0.16 | 2.77 |
| 2022 | 0.38 | 0.51 | 0.41 | 0.37 | 0.42 | 0.78 | 0.13 | 4.18 |
| Based on PPP USD |  |  |  |  |  |  |  |  |
| 2018 | 0.28 | 0.37 | 0.20 | 0.26 | 0.56 | 0.59 | 0.13 | 2.23 |
| 2019 | 0.27 | 0.38 | 0.23 | 0.28 | 0.53 | 0.60 | 0.12 | 2.36 |
| 2020 | 0.26 | 0.41 | 0.23 | 0.30 | 0.46 | 0.59 | 0.12 | 2.09 |
| 2021 | 0.26 | 0.45 | 0.25 | 0.31 | 0.41 | 0.50 | 0.13 | 2.07 |
| 2022 | 0.26 | 0.45 | 0.34 | 0.36 | 0.36 | 0.45 | 0.10 | 2.69 |

Base country: Korea; PPP: purchasing power parity.

other hand, the U.K and the U.S. witnessed a substantial increase in the index (the U.K. from 0.43 in 2018 to 0.60 in 2022; the U.S. from 2.38 in 2018 to 3.04 in 2022).

## Antibiotics

For antibiotics, all countries but France, Italy, and Japan have the index that is greater than one (Table 5). Unlike other drug classes, antibiotics saw increases in the price index from 2018 to 2022 in many countries including France (from 0.78 to 0.87), Italy (from 0.80 to 0.87), the U.K. (from 1.35 to 1.55), Germany (from 1.78 to 1.81), and the U.S. (from 2.86 to 3.34). In only two countries, the price index decreased (from 1.85 to 1.73 for Canada, from 0.74 to 0.71 for Japan).

In general, the price indices are lowest for lipid-modifying agents and highest for antibiotics among the four therapeutic classes (Fig 1).

**Table 4. Laspeyres price indices for antihypertensive drugs, 2018–2022.**

| | Canada | France | Germany | Italy | Japan | Switzerland | U.K. | U.S. |
|---|---|---|---|---|---|---|---|---|
| Based on USD | | | | | | | | |
| 2018 | 0.59 | 0.39 | 0.72 | 0.45 | 0.70 | 1.03 | 0.50 | 3.06 |
| 2019 | 0.60 | 0.38 | 0.66 | 0.46 | 0.73 | 1.04 | 0.63 | 3.59 |
| 2020 | 0.59 | 0.39 | 0.67 | 0.47 | 0.65 | 1.08 | 0.69 | 3.73 |
| 2021 | 0.61 | 0.36 | 0.66 | 0.43 | 0.54 | 1.06 | 0.71 | 3.77 |
| 2022 | 0.64 | 0.36 | 0.68 | 0.41 | 0.43 | 1.15 | 0.79 | 4.71 |
| Based on PPP USD | | | | | | | | |
| 2018 | 0.49 | 0.34 | 0.64 | 0.43 | 0.58 | 0.66 | 0.43 | 2.38 |
| 2019 | 0.48 | 0.35 | 0.60 | 0.45 | 0.56 | 0.66 | 0.53 | 2.64 |
| 2020 | 0.46 | 0.34 | 0.57 | 0.45 | 0.48 | 0.63 | 0.56 | 2.64 |
| 2021 | 0.46 | 0.32 | 0.57 | 0.42 | 0.44 | 0.66 | 0.57 | 2.81 |
| 2022 | 0.44 | 0.31 | 0.57 | 0.40 | 0.38 | 0.67 | 0.60 | 3.04 |

Base country: Korea; PPP: purchasing power parity.

**Table 5. Laspeyres price indices for antibiotics, 2018–2022.**

| | Canada | France | Germany | Italy | Japan | Switzerland | U.K. | U.S. |
|---|---|---|---|---|---|---|---|---|
| Based on USD | | | | | | | | |
| 2018 | 2.22 | 0.89 | 1.99 | 0.82 | 0.89 | 2.49 | 1.59 | 3.68 |
| 2019 | 2.40 | 0.89 | 2.00 | 0.82 | 0.94 | 2.41 | 1.67 | 3.77 |
| 2020 | 2.39 | 0.96 | 2.17 | 0.88 | 0.95 | 2.59 | 1.91 | 3.42 |
| 2021 | 2.46 | 0.97 | 2.23 | 0.88 | 0.87 | 2.60 | 2.10 | 4.15 |
| 2022 | 2.53 | 1.00 | 2.16 | 0.89 | 0.82 | 2.80 | 2.03 | 5.18 |
| Based on PPP USD | | | | | | | | |
| 2018 | 1.85 | 0.78 | 1.78 | 0.80 | 0.74 | 1.61 | 1.35 | 2.86 |
| 2019 | 1.90 | 0.82 | 1.81 | 0.82 | 0.73 | 1.53 | 1.41 | 2.77 |
| 2020 | 1.84 | 0.84 | 1.86 | 0.85 | 0.70 | 1.52 | 1.54 | 2.43 |
| 2021 | 1.86 | 0.85 | 1.91 | 0.86 | 0.70 | 1.60 | 1.68 | 3.10 |
| 2022 | 1.73 | 0.87 | 1.81 | 0.87 | 0.71 | 1.64 | 1.55 | 3.34 |

Base country: Korea; PPP: purchasing power parity.

## Discussion

This present study shows that there are wide variations in the price levels of generic drugs across high-income countries. This study further indicates that the prices of generic drugs are higher in Korea than in other developed countries for some therapeutic categories, such as lipid-modifying agents and antihypertensive drugs. This finding suggests that there is a potential for reductions in the prices of generic drugs in Korea. Drug prices are influenced by a number of factors including market dynamics and government policies regarding pricing, reimbursement, tendering, and purchasing [16]. In general, pharmaceutical prices are known to be higher in countries that have a free market approach to pricing than their counterparts [17]. For example, setting the prices of pharmaceuticals in the U.S. relies primarily on market forces [18]. The prices of prescription drugs in the U.S. are more than two times the prices in

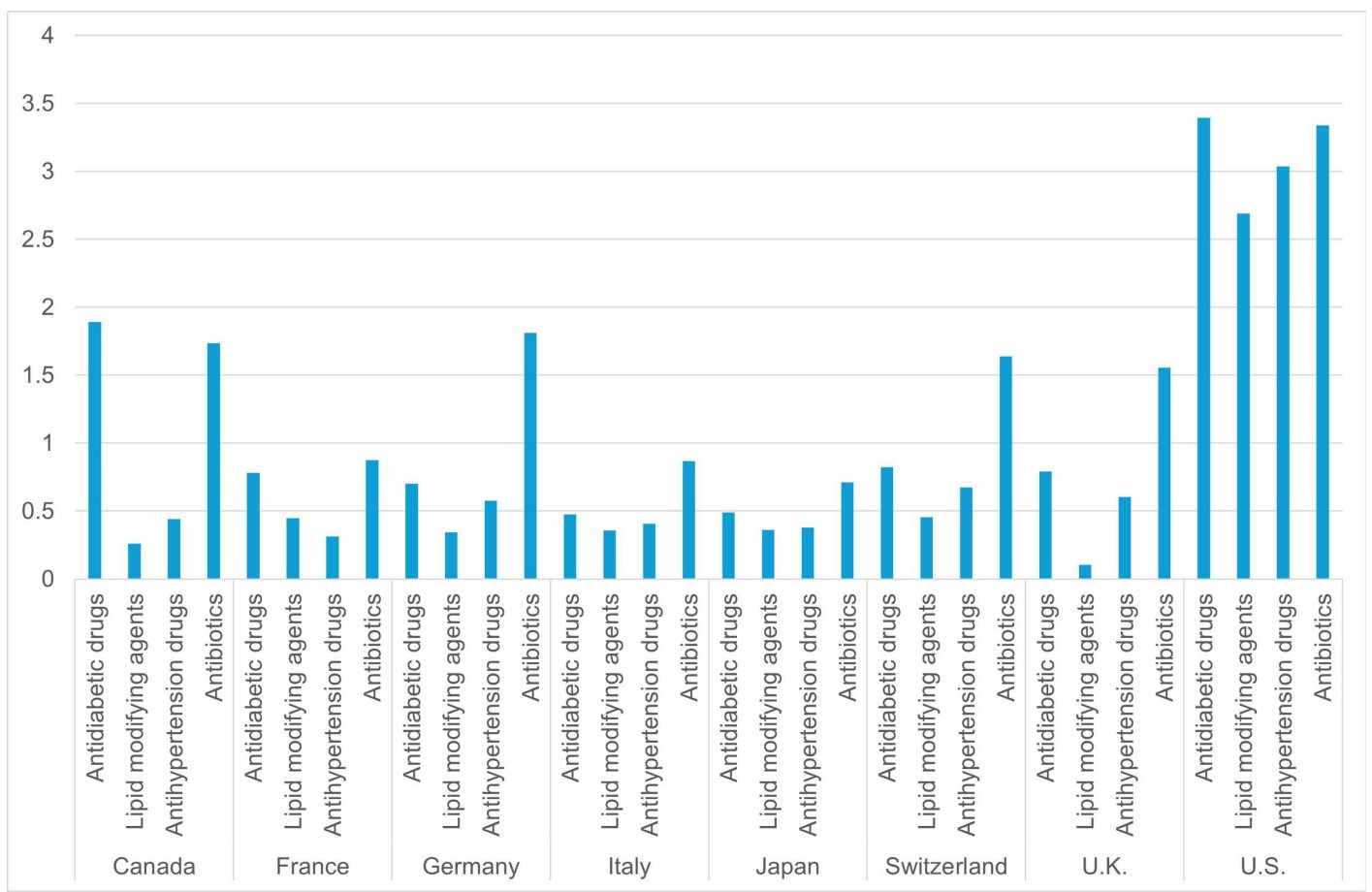

**Fig 1. Laspeyres price indices for generic drugs by therapeutic class in 2022 (base country: Korea, PPP USD).**

other high-income countries [8]. Moreover, the prices of generic drugs are linked to the number of manufacturers in the drug category [19]. According to a U.S. study, the generic drug market that is more competitive witnessed a greater decrease in prices over time [20]. Similarly, an investigation of the European pharmaceutical markets shows that markets with larger share of generics tend to see deeper cuts in drug prices than other countries [21]. The market that has a greater number of generic drugs shows a gradually yet consistently downward trend in generics' share of the total pharmaceutical expenditure in Korea [3]. The findings of these existing studies suggest that high prices of generics in Korea might reflect limited competition in the generics market.

The generic price differences might stem from regulatory regimes related to pharmaceutical pricing. Many European countries adopt a variety of mechanisms to regulate generic prices, such as reference pricing and price cap [22]. However, questions have been raised regarding the effectiveness of these price control policies in highly-regulated markets like Europe, since the interventions fail to incentivize for further voluntary reductions above and beyond what is required, therefore hampering price competition [22]. Likewise, a series of measures implemented in Korea to control generic drug prices do not appear to have a material impact on pharmaceutical spending. A more effective strategy to promote price competition would be to combine price control mechanisms with a policy measure to encourage use of lower-priced generics as they enter the market.

In our analysis, there are wide variations in the temporal trend of generic price indices across therapeutic groups. In general, the price indices of generic antihypertensive drugs and antibiotics in eight high-income countries in comparison to Korea have increased from 2018 to 2022. On the other hand, the price indices of generic antidiabetic and lipid-modifying drugs have decreased during the same period. As a result, although comparison countries paid, on average, almost twice what Korea did for generic antidiabetics in 2018, the gap has diminished over time. On the other hand, other countries paid 60% of the prices for generic lipid-modifying drugs that Korea paid in 2018 and paid even less (52%) in 2022.

For antibiotics, all countries but France, Italy, and Japan have the index that is greater than one, indicating that the prices of generic antibiotics were higher in many high-income countries than in Korea. Nevertheless, the prices of generic antibiotics increased in many countries including Germany, the U.K., and the U.S. whose price indices were greater than one. Overall, the price index for generic antibiotics has increased in all of comparison countries combined during the study period. Sharp increases in the prices of generic antibiotics in the past in the U.S were attributed to reduced competition in terms of the number of manufacturers [23]. In recent years, shortages of antibiotics that are prescribed to treat common infections and available in generic formations have raised concerns in some Western countries [24]. According to a study in the Canadian pharmaceutical market, the segment of the market that has only one generic manufacturer is more likely to face a drug shortage than other segments with multiple manufacturers [25]. Prices of generic drugs are linked to their shortages such that lower-priced generic drugs in the U.S. are at an increased risk of shortage [26]. Manufacturing disruptions have been considered one of the key factors contributing to drug shortages worldwide [27]. The COVID-19 pandemic posed a burden on the pharmaceutical supply chain and was viewed to have contributed to increasing prices of some essential drugs like analgesics in some countries [28,29]. One way to address a shortage of a drug was to increase the price of the drug [30,31]. U.S. data indicate that the country has experienced more pharmaceutical price increases in 2021 and 2022 than in the preceding four years [32]. Likewise, drugs in short supply tend to see higher price hikes in China than other drugs between 2019 and 2022 [33].

Generic drug prices for lipid-lowering and antihypertensive drugs in South Korea are higher compared to other high-income countries. In comparison, Korea has benefitted from lower prices of generic antibiotics and lipid-modifying agents. The findings of this study suggest that there is a potential for price reductions in some therapeutic classes. However, it should be recognized that international comparison based on the volume-weighted price indices are reliant upon the consumption pattern of drugs. That is, the changes observed in this present study reflect changes not only in the prices of generic drugs but also in the mix of drugs used in the base country, Korea. If more expensive generics are preferred in Korea, the results of the analyses might point toward the underestimation of price indices.

Overall, generic prices as a proportion of originator prices in Korea are known to be higher than in other high-income countries [34]. Specifically, generic prices in Korea are approximately 50% of originator prices, whereas in many European countries, they range from 2% to 10%. However, in this study, generic prices for lipid-modifying agents and antibiotics are found to be lower in Korea compared to other high-income countries. It is possible that originator prices are lower in Korea for these specific drug classes. So far, generic price reductions in Korea have been determined proportionally to the originator prices. However, the findings of this current study suggest that a more targeted approach to generic price cuts should be considered. It is recommended to target price cuts for specific drug classes, such as lipid-modifying agents and antihypertensives, where the prices of generic drugs in Korea are higher than those in other high-income countries.

The findings of this study should be interpreted with caution. First, the results of drug price indices may have been influenced by a number of methodological considerations, such as the index chosen, currency conversion, and kind of prices analyzed [11,35]. Second, there is a wide variation in the range of active ingredients, dosage forms, and strengths available across countries, which made selecting a representative sample of generic drugs difficult [36]. Third, we based our analyses on the price charged to the pharmacy but did not account for rebates, which are substantial and required in some countries [32]. Net prices account for rebates paid to insurance plans and pharmacy benefit managers, which appear to be growing in the U.S. in recent years [32] and yet research based on net prices is sparse [16]. Use of net prices would have estimated the price indices lower than this study finds.

## Conclusion

This study shows that the prices of generic drugs are higher in Korea than in other high-income countries for lipid-modifying agents and antihypertensive drugs. This study suggests adopting a more targeted approach, particularly for drug classes like lipid-modifying agents and antihypertensives, where generic prices are higher than in other high-income countries. To promote price competition, policy interventions to encourage use of cheaper generics should be considered in combination with price control mechanisms. This study further shows that there are wide variations in the temporal trend of generic prices across therapeutic groups. The prices of generic antibiotics are higher in many comparison countries and still have increased from 2018 to 2021.

## Supporting information

**Supplementary Table 1. Study drugs.**
(DOCX)

## Author contributions

**Conceptualization:** Chanmi Park, Dong-Sook Kim.

**Data curation:** Chanmi Park.

**Formal analysis:** Chanmi Park.

**Funding acquisition:** Dong-Sook Kim.

**Investigation:** Chanmi Park.

**Methodology:** Dong-Sook Kim.

**Software:** Chanmi Park.

**Supervision:** Dong-Sook Kim.

**Visualization:** Dong-Sook Kim, Inmyung Song.

**Writing – original draft:** Dong-Sook Kim, Inmyung Song.

**Writing – review & editing:** Chanmi Park, Dong-Sook Kim, Inmyung Song.

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
