## [Decision Letter · Decision Letter 0]

11 Oct 2024

PONE-D-24-21235International comparison of generic drug prices in four therapeutic classesPLOS ONE

Dear Dr. Song,

Thank you for submitting your manuscript to PLOS ONE. After careful consideration, we feel that it has merit but does not fully meet PLOS ONE’s publication criteria as it currently stands. Therefore, we invite you to submit a revised version of the manuscript that addresses the points raised during the review process.

We look forward to receiving your revised manuscript.

Kind regards,

Ebenezer Wiafe, PhD, MPharm, Pharm D

Academic Editor

PLOS ONE

2. Thank you for stating the following financial disclosure: This study was supported by the Ministry of Health and Welfare of Korea (Ref. No. 2023080955F – 00).  

3. Thank you for stating the following in the Acknowledgments Section of your manuscript: This study was supported by the Ministry of Health and Welfare of Korea (Ref. No. 2023080955F – 00). 

Please remove any funding-related text from the manuscript and let us know how you would like to update your Funding Statement. Currently, your Funding Statement reads as follows: This study was supported by the Ministry of Health and Welfare of Korea (Ref. No. 2023080955F – 00).

5. We notice that your supplementary tables are included in the manuscript file. Please remove them and upload them with the file type 'Supporting Information'. Please ensure that each Supporting Information file has a legend listed in the manuscript after the references list.

Reviewers' comments:

Reviewer's Responses to Questions

**Comments to the Author**

1. Is the manuscript technically sound, and do the data support the conclusions?

Reviewer #1: Partly

Reviewer #2: Yes

2. Has the statistical analysis been performed appropriately and rigorously? 

Reviewer #1: N/A

Reviewer #2: Yes

3. Have the authors made all data underlying the findings in their manuscript fully available?

Reviewer #1: Yes

Reviewer #2: Yes

4. Is the manuscript presented in an intelligible fashion and written in standard English?

Reviewer #1: Yes

Reviewer #2: Yes

5. Review Comments to the Author

Reviewer #1: Dear Author,

I think this article covers a very interesting topic related to generic drug prices in South Korea. However, I would like to make a few comments about your manuscript. Follows are the details.

1. Your research is to compare the price of a generic drug in South Korea with eight other countries (Base country: Korea). I recommend that you revise the title to reflect it.

2. Explain in more detail the rationale for choosing 4 therapeutic classes and 8 comparison countries for price comparison of generic drugs.

3. You selected eight countries as comparison countries for generic drug prices, which are considered as reference when determining prices in Korea. Does Korea consider the prices of the eight countries as a reference when determining the price of generic drugs?

4. Despite the lower prices of generic drugs in Korea in some therapeutic classes, can we say that overall the prices of generic drugs in Korea are higher?

- I recommend that you revise the sentences in the Discussion and Conclusion sections related to this issue or provide sufficient rational.

5. Consider modifying the Table 2,3,4,5 to include both USD base and PPP USD base as shown in Table 1.

6. The price indices in Figure 1 PPP USD based?

7. Please make sure that manuscript was written according to the journal's guidelines as a whole. In particular, check the following:

• Please check that all abbreviations present are commonly used and all are defined in their first instance in the manuscript.

• Please check that all abbreviations present in Table/Figure legends. These should also be redefined in the legends if the abbreviation is used.

• Please make sure that all figures, tables and boxes are clearly and correctly titled, described and cited in the text.

• Please add additional detail to figure legends where relevant to ensure all figures are adequately described and appropriate context is given.

Reviewer #2: The discussion should take into account and elaborate on the impact of drug prices, the locations of drug production, and the risk of shortages. Indeed, it is known (and observed in this work) that drug prices in France are often very low, but it also appears that supply disruptions in France are very frequent. The proximity of South Korea to Asian production lines could allow the country to prevent such problems, but this needs to be discussed further.

6. PLOS authors have the option to publish the peer review history of their article (what does this mean? ). If published, this will include your full peer review and any attached files.

**Do you want your identity to be public for this peer review?** For information about this choice, including consent withdrawal, please see our Privacy Policy .

Reviewer #1: No

Reviewer #2: No

---

## [Decision Letter · Decision Letter 1]

6 Feb 2025

Comparison of Generic Drug Prices in Korea and Eight High-Income Countries Across Four Therapeutic Classes

PONE-D-24-21235R1

Dear Dr. Song,

We’re pleased to inform you that your manuscript has been judged scientifically suitable for publication and will be formally accepted for publication once it meets all outstanding technical requirements.

Kind regards,

Ebenezer Wiafe, PhD, MPharm, Pharm D

Academic Editor

PLOS ONE

Additional Editor Comments (optional):

Reviewers' comments:

Reviewer's Responses to Questions

**Comments to the Author**

1. If the authors have adequately addressed your comments raised in a previous round of review and you feel that this manuscript is now acceptable for publication, you may indicate that here to bypass the “Comments to the Author” section, enter your conflict of interest statement in the “Confidential to Editor” section, and submit your "Accept" recommendation.

Reviewer #1: All comments have been addressed

Reviewer #2: All comments have been addressed

2. Is the manuscript technically sound, and do the data support the conclusions?

Reviewer #1: Yes

Reviewer #2: Yes

3. Has the statistical analysis been performed appropriately and rigorously? 

Reviewer #1: Yes

Reviewer #2: Yes

4. Have the authors made all data underlying the findings in their manuscript fully available?

Reviewer #1: Yes

Reviewer #2: Yes

5. Is the manuscript presented in an intelligible fashion and written in standard English?

Reviewer #1: Yes

Reviewer #2: Yes

6. Review Comments to the Author

Reviewer #1: (No Response)

Reviewer #2: Comments were taken into account and manuscript modified as recommanded.It is now clearer and more precise (i.e manuscript title)

7. PLOS authors have the option to publish the peer review history of their article (what does this mean? ). If published, this will include your full peer review and any attached files.

**Do you want your identity to be public for this peer review?** For information about this choice, including consent withdrawal, please see our Privacy Policy .

Reviewer #1: No

Reviewer #2: No

---

## [Editor Report · Acceptance letter]

PONE-D-24-21235R1

PLOS ONE

Dear Dr. Song,

I'm pleased to inform you that your manuscript has been deemed suitable for publication in PLOS ONE. Congratulations! Your manuscript is now being handed over to our production team.

Kind regards,

on behalf of

Dr. Ebenezer Wiafe

Academic Editor

PLOS ONE